# The Possibility of Using Sulphur Shelf Fungus (*Laetiporus sulphureus*) in the Food Industry and in Medicine—A Review

**DOI:** 10.3390/foods12071539

**Published:** 2023-04-05

**Authors:** Iwona Adamska

**Affiliations:** Department of Fish, Plant and Gastronomic Technology, West Pomeranian University of Technology in Szczecin, 70-310 Szczecin, Poland; iwona.adamska@zut.edu.pl

**Keywords:** sulphur shelf fungus, mushroom, chicken of the woods, phytochemicals, health-promoting properties, food, *Laetiporus sulphureus*

## Abstract

Sulphur shelf fungus (*Laetiporus sulphureus*) has so far been largely underestimated as a potential raw material for the food industry. Many studies have demonstrated that the extracts obtained from this mushroom and some of their components have positive effects on human health. They have antioxidant, antibacterial, and anticancer properties and regulate human metabolism and digestive processes. Water extracts also have this effect. In addition, the substances contained in this mushroom have the ability to preserve food by inhibiting the growth of undesirable microorganisms. These properties have led to the situation that in some countries, shelf sulphur fungus is legally recognized as a raw material that meets the requirements of the food and processing industries. This paper is a review of the latest information (mainly for the period 2016–2023) on the chemical composition and the possibility of using *L. sulphureus* in the food industry and in medicine.

## 1. Introduction

Sulphur shelf fungus (*Laetiporus sulphureus*) is a parasitic fungus with a characteristic yellow color and inhabits older trees. It occurs on all continents of the globe, but only in some regions is it used for culinary purposes. Due to its color, it is called “chicken of the woods” in the US. The fruiting bodies are a source of many substances valuable in culinary and medical fields. Due to the known method of cultivation of this mushroom, together with its rich chemical composition and beneficial effects for the human body, it is worth considering as a potential raw material for industry. The aim of this paper is to review the latest information (from 2016 to 2023) on the chemical composition of *L. sulphureus* fruiting bodies and the possibilities for their use in food production and in medicine.

## 2. Characteristics and Occurrence of *Laetiporus sulphureus*

*Laetiporus sulphureus* is a fungus belonging to the phylum Basidiomycota, family Polyporaceae. Its fruiting bodies are semicircular, up to 40 cm in diameter, with a short stem. They grow on tree trunks, are console-shaped and often overlap (grow in cascades). The bright, sulphur yellow to vivid yellow color is characteristic only of young fruiting bodies. The older fruiting bodies are strongly discolored, being light orange or white-orange, sometimes with brown spots. With age, they become dry, hard and compact (less brittle) (Figure 1). On the underside of the fruiting body is a layer of tubular hymenophores [1,2].

This fungus occurs naturally as a parasite on a variety of trees, causing brown rot (rot and holes form in the wood), which leads to their rapid death. Most often it inhabits deciduous trees (mainly of the genera *Quercus*, *Robinia* and *Populus*), and much less often, coniferous trees [3,4]. The infection of trees occurs as a result of their injury or through unprotected wounds that arise during maintenance treatments (crown adjustment) or natural damage to trees. The fruiting bodies are formed most often high on the trunk and less often on felled trees, just above the ground [5].

**Occurrence.** *Laetiporus sulphureus* is very common: most reports come from North America and Europe [6], but it has also been found in South America, Africa, Asia (including China, Laos and Thailand) and Australia [3,4,5,7,8]. It occurs in parks, forests and woodlots up to 2000 m above sea level [5,7].

**Cultivation on culture media and in home conditions.** *Laetiporus sulphureus* can also be cultivated. The type and composition of the substrate affects the appearance of the mycelium [9], the growth rate of the fungus [9,10,11,12], and thus also the formation of fruiting bodies. The best composition of the substrate and conditions for its cultivation have been determined, which make it possible to obtain fruiting bodies as early as 46–52 days after the establishment of the cultivation [13]. The growing medium should be sufficiently compact and densely packed so that the *L. sulphureus* mycelium can easily grow out of it. Therefore, a mixture, for example, of sawdust and straw, is a better substrate than a homogeneous substrate composed only of straw [11]. In the case of a substrate composed of wheat bran, the factor conducive to better growth of the mycelium was the addition of alder, larch and oak sawdust [9].

The process of formation of fruiting bodies can be induced. According to Pleszczyńska et al. [14], an effective method is to shock the mycelium with water or low temperature in winter. However, this failed in the case of cultivation conducted by Simoes [11] on wheat grains and various types of production substrates, despite testing various methods of inducing the formation of fruiting bodies. Although during continuous induction with low temperature, the mycelium formed structures similar to *L. sulphureus* fruiting bodies, they were not typically shaped fruiting bodies. The author of the research, however, stated that the reason for this may have been the difficulty in maintaining a constant high level of humidity.

Cultures for obtaining biomass of *L. sulphureus* mycelium intended for food purposes or for the extraction of selected substances (e.g., laetiporic acid) can be grown on nutrient media (mycelial cultures). Media solidified with agar are the most commonly used in the studies, e.g., PDA (potato dextrose agar) without additives or enriched with additional ingredients [9,10,11], MEA (malt extract agar) [10] or liquid media, e.g., PDB [11,12]. The best substrates were PDA media [9,10] enriched with four combinations of additives: (1)—dehydrated potato infusion and dextrose, (2)—malt and yeast extracts, glucose and peptone, (3)—malt extract, peptone and xylitol and (4)—extracts from beech sawdust and malt, peptone and xylose [9]. MEA with the addition of sorghum grains is also a valuable substrate. However, in the case of using other seeds (for example, pearl millet and barley), the addition of gypsum and CaCO_3_ is recommended [10]. According to Jasińska et al. [15], it is possible to grow this fungus on an agar medium with the addition of residues from biogas production (digestate), which promotes faster formation of fruiting bodies compared with other types of tested media. The optimal growth conditions are temperatures of 25–30 °C and pH 6–8 [10].

To date, the cultivation of this fungus has usually been carried out on a smaller scale, and the purpose was mainly to obtain raw material for obtaining enzymes. For culinary purposes, fruiting bodies occurring in the wild (in natural conditions) are used. However, home cultivation of *L. sulphureus* is now beginning to spread. Starters containing inoculum in liquid form [16] or solid (inoculum carrier is sawdust) [17] ready for direct application are available in online stores. The inoculum package comes with application and cultivation instructions. In home cultivation, the growing medium is usually a log of freshly cut hardwood, e.g., oak or ash. The inoculum should be applied on the planes of intersection of the wood or into drilled holes. After 2–3 months of incubation, the cultures are partially dug into the soil or left on the soil surface and covered with a thin layer of wood shavings. Fruiting bodies are obtained one year after inoculation of the substrate [16,17], and the harvesting is carried out for several subsequent years.

Several studies have investigated the genome of *L. sulphureus*. Two research teams working independently have successfully sequenced the genome of the fungus *L. sulphureus* [18,19]. One of the tested mushrooms came from the UK. During the study, the genome assembly was found to be 37.4 megabases long and made up of 14 chromosomes [19]. The second of the tested isolates (NWAFU-1) came from a natural site in China. During the study, it was found that there is variation within the genome between the subspecies of the fungus. Genes have been identified that encode the synthesis of carbohydrates, polysaccharides and secondary metabolites, i.e., substances responsible for the bioactivity of the fungus (e.g., eburicoic acid, trametenolic acid and its derivatives, laetiporic acid A, sulfurenic acid, and dehydrosulfurenic acid) and ergosteroids (e.g., ergost-5,7,22-trien-3-ol or ergosta-7,22-diene-3,5,6-triol). Sixteen compounds from the ergosteroids and lanostanoids groups were identified, of which seven were found only in fruiting bodies, eight only in the mycelium, and one (ergost -3,5,7,9(11),22-pentaen) in both structures: mycelium and fruiting bodies. In addition, *L. sulphureus* has been shown to have a tetrapolar mating system. Comparative studies of *L. sulphureus* strains have shown that the subspecies differ in terms of the ability to synthesize many different secondary metabolites, including terpenoids and polysaccharides. In the case of the analyzed isolate (NWAFU-1), this activity is high. The results of these studies give the opportunity to select the most valuable strains for obtaining substances of medical and economic importance [18].

## 3. Nutritional Value of Fruiting Bodies

The fungus *Laetiporus sulphureus* forms fruiting bodies on tree trunks, but its mycelium develops in the wood of the host plant and draws water, mineral salts and other substances from there. Due to the biology of the sulphur shelf fungus, and in particular the large variety of plant species on which it occurs, the chemical composition of this fungus is very rich (Table 1), and at the same time, it may slightly differ. The caloric value of *Laetiporus sulphureus* fruiting bodies is in the range 321.7–375.6 kcal dw [1,8,20,21,22], and the water content is 66.7–91.5 g/100 g of the fruiting body (Saha et al. [23] and Florczak et al. [24], respectively). Among the basic components of the dry matter, carbohydrates dominate, which constitute 64.9–74.5% of dw. The content of the other components is much lower: protein 8.6–21.0% dw; fat 2.3–5.9% dw; ash 4.0–9.0% dw; and fiber 4.1–15.2% of dry matter [1,8,20,21,22,23,24,25] (Figure 2). The largest fraction of neutral lipids were triglycerides [8].

**Protein.** The fruiting body protein has a high biological value with a high content of digestible protein (86.1% of crude protein) [1]. These mushrooms contain free amino acids beneficial for the human body with a content from 3.63 mg g^–1^ according to Turfan et al. [31] to 17.12 mg g^−1^ dw (i.e., 16.2% of crude protein) according to Kovacs and Vetter [1]. The protein is characterized by a composition of protein fractions typical of fungi, which is different from the distribution observed in plant materials. In the study of Kovacs and Vetter [1], the protein fraction included a high content of albumins (51.3% of crude proteins on average) and globulins (11.1%), and a low proportion of prolamine and prolamine-like substances (they accounted for a total of 7.3% of the crude protein on average). In addition, they reported a relatively high amount of glutelins and glutelin-like substances (30.3%).

These fruiting bodies contain about 3.25 g/100 g dw of total nitrogen, including 76.5% of protein nitrogen and a total of 30.7% of non-protein nitrogen (27.1% in water-soluble compounds and 3.6% in water-insoluble compounds) [24]. Turfan et al. [31] reported a total soluble protein content of 83.3 mg g^–1^ dw. The high proportion of albumins and globulins affects the speed and ease of digestion of these mushrooms and the absorption of the nutrients they contain. A similar share of individual protein fractions in the protein of *L. sulphureus* fruiting bodies was obtained by Petrowska [70]. Discrepancies in the content of individual components and fractions may result from the diversity of environmental conditions in which these fungi grew, and especially the species of the plant that hosts the fungus, as demonstrated by Kovacs and Vetter [1]. Bulam et al. [8], based on the analysis of the research results of Agafanova et al. [36], pointed out that slight differences in the composition of amino acids, especially essential amino acids, could also depend on the fungus strain (the size of the difference is usually below 0.5%). The greatest difference was found in the strains LS-BG-0804 and LS-UK-0704 for glutamic acid levels (0.9% and 0.4%, respectively) and tyrosine (0.3% and 0.8%, respectively). The fruiting bodies of *L. sulphureus* have also been shown to contain exogenous arginine, histidine, isoleucine, leucine, methionine, threonine and tryptophan, with minimal differences in their content in the two strains of fungus [36]. Wang [45] also showed the presence of aspartic acid, serine, glycine, alanine, cysteine, valine, phenylalanine, lysine and proline in the extract.

**Fatty acids.** Many studies have shown a favorable profile of fatty acids found in *L. sulphureus*; however, the amounts of individual fatty acids are variable. Oleic acid (C18:1) predominates, followed by palmitic (C16:0) and stearic (C18:0) acids, and there are significantly smaller quantities of linoleic (C18:2), myristic (C14:0) and palmitoleic (C16:1) acids [34]. Similar results were obtained by Palazzolo et al. [37]: oleic acid dominated in fruiting bodies, there was less linoleic acid, and palmitic and stearic acids had the smallest share. Bengu et al. [34] found that compared with other fungal species tested (*Suillus luteus* and *Corrinus atramentarius*), *L. sulphureus* fruiting bodies contained the most myristic acid, palmitic, steraic and oleic acids.

Based on the analysis of the results obtained by Agafanova et al. [36], Bulam et al. [8] showed a high level of unsaturated fatty acids in relation to saturated fatty acids (UFA/SFA > 3.4). In the fruiting bodies of *L. sulphureus* studied by them, linoleic acid was the most abundant, and there were slightly smaller quantities of oleic and palmitic acids [36]. Similar results were obtained by Petrovic et al. [21], Sinanoglou et al. [38] and Woldegiorgis et al. [35].

**Carbohydrates.** *Laetiporus sulphureus* fruiting bodies contain 57.3 mg g^−1^ dm total soluble saccharides. These mushrooms contain higher amounts of soluble oligosaccharides (56.0% total soluble saccharides; 32.1 mg g^−1^ dm) than soluble polysaccharides (44%; 25.2 mg g^−1^ dm) [1]. The content of water-soluble polysaccharides depends on the age of the fruiting bodies: a higher share of these substances was found in mature mushrooms, and the smallest share in aging mushrooms. The opposite pattern was observed in the case of base-soluble polysaccharides which were the most abundant in aging fruiting bodies [51]. Alkaline-soluble polysaccharides [71] and water-soluble endopolysaccharides (glucans, galactans, and glycoproteins) were also isolated from these fungi. *Laetiporus sulphureus* fruiting bodies contain arabinose, galactose, glucose, xylose, mannose, rhamnose and fucose [26]. In addition, the analysis of the composition of the post-culture medium showed the presence of the monosaccharides fucose, arabinose, xylose, mannose, galactose and glucose [27]. The fruiting bodies also contained six laetiporans (polysaccharides, which are marked with the letters A–F). Their content was different in individual fractions of the extract, but generally laetiporan A was the most abundant, although it constituted only 0.28% of the fruiting body weight, and laetiporan F had the lowest share (0.06%) [26]. Among the polysaccharides present in the extract from this mushroom, laminaran (β-glucan) and fucomannogalactan were also recognized [72].

*Laetiporus sulphureus* has one of the highest values of total soluble carbohydrates among the 15 compared mushroom taxa (266.8 mg g^−1^). Only *Craterellus cornucopioides*, *Hericium erinaceus*, *Morchella conica*, one of the strains of *Pleurotus ostreatus* and *Lactarius deliciosus* had a higher content. In addition, this mushroom also contained high levels of glucose (40.5 mg g^−1^), while the content of fructose was at an average level (6.3 mg g^−1^), and sucrose was one of the lowest (0.32 mg g^−1^) among the compared species. The high content of soluble carbohydrates and sugars in mushrooms causes their slightly sweetish taste. This feature was found to be desirable in mushrooms [31].

The quantity of glucans present in the sulphur shelf fungus and their type depends on the method of preparation of the tested extract. In aqueous extracts and extracts of partially purified polysaccharides, β-glucans dominated (they accounted for 92.0% and 90.8% of glucans present in the extract, respectively), while in extracts of hot alkaline extracted polysaccharides, α-glucans (67.1%) dominated. A positive correlation has been shown between the amount of α-glucan and antioxidant activity [49]. Polysaccharides from the group of α-(1→3)-glucans are substances of great importance for dentistry. Depending on the extraction method used, Wiater et al. [73] obtained from 32.1 to 56.9% of this fraction from the dry matter of the mycelium. The content of α-(1→3)-glucans in the walls of the sulphur shelf fungus depended on the development stage of the fruiting body: it was the lowest in young, immature fruiting bodies (17.3–37.6%), and the highest in mature fruiting bodies (bodies with mature spores; 42.8–47.8) [74]. In further research, Wiater et al. [75] isolated a water-soluble group of α-(1→3)-glucooligosaccharides from α-(1→3)-glucan and demonstrated its high probiotic importance for the human body.

The content of free carbohydrates changes depending on the age of the fruiting body: the lowest was in young mushrooms (13.9%), and the highest in aging mushrooms (27.2%). The mannite content also increased with the age of the fungus: it was 9% in young fruiting bodies and 25.2% in aging fruiting bodies [51]. According to Petrovic et al. [21], in the group of free sugars, trehalose (4.0 g/100 g dw) dominates, then mannitol (3.5 g/100 g dw), and fructose (0.5 g/100 g dw) is the least abundant. In addition, there are small amounts of glucose and sucrose in this mushroom [8]. Chitin content also changes with age: the lowest amount was found in mature mushrooms (2%), and the highest in aging mushrooms (4.9%) [51]. However, it should be recognized that this component is present in the fruiting bodies of *L. sulphureus* in very small amounts. Florczak et al. [24] reported it at a level of 0.1 g/100 g dw, and this value was very low compared with the amounts found in another mushroom species compared in their studies (e.g., fruiting bodies of *Flammulina velutipes* contained 8.2 g/100 g dw of chitin).

Based on the comparison of the nutritional values of selected edible mushrooms, it was found that *L. sulphureus* is the species that is the most protein-poor, but the most carbohydrate-rich. The other main components (fat, ash and fiber) are present in amounts similar to those found in other species of mushrooms (Figure 3).

**Mineral elements.** The content of mineral salts in mushrooms depends on the habitat in which the mushroom developed [8]. Due to the fact that *L. sulphureus* is an arboreal fungus, the host tree species and at the same time, the habitat of the fungus are important. Therefore, the nutritional status and health of the plant play an important role. Diseases of the host plant, both of bacterial and fungal etiology, can significantly affect the content of minerals in the fungus, and thus indirectly affect the size and rate of formation of fruiting bodies. Bulam et al. [8], based on the results of research conducted by, among others, Agafonova et al. [36], Ayaz et al. [20], Palazzolo et al. [37], Luangharn et al. [25], Saha et al. [23], Kovacs and Vetter [1], Turfan et al. [31] and Bengu [34], found that there is a relationship between the mineral content in the fruiting bodies of *L. sulphureus* and the time of mushroom harvesting, methods of cultivation and the adopted method of determining the content of a given component. Differences in the contents of particular mineral elements are shown in Table 2. However, according to research by Kovacs and Vetter [1], fruiting bodies of *L. sulphureus* have a similar mineral composition to other fungi growing on trees. They found that 99% of the total mineral content of this mushroom consists of potassium, phosphorus, calcium and magnesium. In addition, they showed that the fruiting bodies contain large amounts of phosphorus and potassium, satisfactory amounts of iron, zinc and copper, and very small amounts of elements toxic to humans: arsenic, chromium and nickel. Bengu [34] showed that the amount of iron in the fruiting bodies of *L. sulphureus* corresponds to the daily RDA dose and suggested that this mushroom could be included in the diet of people suffering from anemia.

**Vitamins and organic acids.** The content of vitamins in the fruiting bodies of *L. sulphureus* has been rarely studied. Among the tocopherols present in these mushrooms, the largest share, as much as 57.6%, consisted of α-tocopherol (its content in 100 g of dry weight of the fruiting body was 109.3 µg), and δ-tocopherol was the smallest share (9.7%; 18.4 µg). The remainder consisted of γ-tocopherol (62.1 µg) [21]. Szymański et al. [2] showed the presence of niacin in the fruiting bodies of *L. sulphureus*. Studies of other mushroom species have shown that these organisms are also a source of pantothenic acid, biotin, vitamins B12, D3 [8], ascorbic acid and β-carotene [81,82,83].

Ayaz et al. [20] determined and compared the content of chemical components in eight species of mushrooms. The fruiting bodies of *L. sulphureus* contained high levels of malic acid (3.7 g/kg), less citric acid (3.1 g/kg) and very little ascorbic acid (0.1 g/kg), but this mushroom had the lowest levels of these components among the compared species. The low content of ascorbic acid (514.2 µg/g dw) in these fruiting bodies was also shown in the research conducted by Acharya et al. [50]. Petrović and colleagues [21] determined the level of citric acid in this mushroom (1.2 g/100 g dw), oxalic acid (2.7 g/100 g dw), fumaric acid (0.3 g/100 g dw) and quinic acid (0.2 g/100 g dw). Sulphur yellow fungus also contains protocatechuic acid (measured at 17.7 µg/g dw [56] and 3.3 mg/100 g dw [47]), gallic acid (16.0 µg/g dw [53] and 0.4 mg/100 g dw [47]), chlorogenic acid (9.7 µg/g dw), caffeic acid (8.4 µg/g dw) and p-coumaric acid (8.0 µg/g dw) [53]. The presence of gallic and protocatechuic acids in extracts from *L. sulphureus* fruiting bodies was also demonstrated by Karaman et al. [55]. Among the phenolic acids and derivatives, the fruiting bodies of *L. sulphureus* also contained p-hydroxybenzoic acid (measured at 30.7 µg/100 g dw [21] and 4.1 mg/100 g dw [47]), cinnamic acid (144.6 µg/100 g dw) [21], kojic acid (0.02 mg/100 g dw) [47] and isovaleric acid [40].

The content of chemical components, including organic acids, in the fruiting bodies of *L. sulphureus* varies depending on the developmental stage of the fungus. The highest total content of organic acids was found in old mushrooms (5.1%), whereas adult mushrooms contained 3.1% and young mushrooms 3.3%. In all age groups of fruiting bodies, tartaric acid was always dominant, followed by malic acid. The greatest differences in the content of individual acids were found in young mushrooms (from 0.1% for succinic acid to 1.4% for tartaric acid). The greatest differences were observed for citric acid; the levels in the aging fruiting body were three times greater than in the young mushroom (1.5% and 0.5%, respectively). The reverse pattern was observed in the case of malonic acid. Its level in young mushrooms was high (1.1%), whereas in adult fruiting bodies it had decreased by almost half (to 0.6%), and in the old specimens, it increased strongly and reached the maximum value (1.3%) [51].

**Other components.** The presence of quercetin (1.1 μg/g d.w.), kaempferol (0.9 μg/g d.w.) and (+)-catechin (4.7 μg/g d.w.) has been reported in *L. sulphureus* fruiting bodies [53]. They also contain sterols, dominated by ergosterol (136.9 mg/100 g dw) and ergosterol peroxide (64.0 mg/100 g dw); L-tocopherol (0.16 mg/100 g dw) had a much smaller share. Indole derivatives are also important components of these mushrooms: relatively high amounts of L-tryptophan were found (14.1 mg/100 g dw), whereas 5-OH-L-tryptophan and tryptamine were present in small amounts (1.5 mg/100 g dw and 1.2 mg/100 g dw, respectively). However, the conducted research did not show the presence of melatonin [47]. An interesting component isolated from *L. sulphureus* is ±-laetirobin, a substance with great potential in medicine, especially in the treatment of cancer [57].

## 4. Bioactivity of *Laetiporus sulphureus*

**Bioactive substances.** Substances responsible for the diverse positive effects of *Laetiporus sulphureus* include tocopherols, steroids, triterpenes, beauvericin, organic acids (including laetiporic and ascorbic acids), lectins, pigments, benzofurans, α-glucans, phenolic compounds, polysaccharides and monosaccharides [1,8,30,33,46,84,85,86]. The composition of bioactive substances found in *L. sulphureus* is similar to the composition of substances found in *Pleurotus ostreatus* [1]. Due to its bioactivity, the sulphur shelf fungus has long been used in traditional medicine [87]. Laboratory tests have proven its positive multidirectional effects on the human body (Figure 4).

**Antioxidant properties of *L. sulphureus*.** Due to their rich and diverse chemical composition, extracts obtained from *L. sulphureus* show antioxidant activity, which is most often assessed as high [79,80,85,88], especially when samples are obtained from dried mushrooms, not mycelium [84]. It is higher than the antioxidant activity of rosemary and coffee acids [80]. However, in some studies, this activity was rated as moderate or even low [89] (e.g., lower than the activity of the BHA control sample [85]). 

The antioxidant activity of *L. sulphureus* extracts is similar to that of extracts obtained from the fruiting bodies of the fungus *Pleurotus ostreatus* [1], but three times higher than in the case of *Trametes versicolor* (free radical-scavenging capacity of the extract determined by the DPPH method; [79]). In another study, *Laetiporus sulphureus* was characterized by the lowest antioxidant activity (determined by the FRAP method) among five fungal species tested (3.5 mmol Trolox/kg dw). The highest activity in this study was found in *Gleophyllum sepiarium* (87.8 mmol Trolox/kg d.w.) (87.8 mmol Trolox/kg dw) [56].

The high antioxidant potential is influenced, among other factors, by the content of phenolic compounds and flavonoids. However, the total phenolic content in extracts prepared with 70% ethanol was considered very low (142.1 mg GAE/l) [89]. The content of polyphenols in water extracts was higher than in ethanol extracts (43.9 and 9.8 mg GAE/100 g DW, respectively) [90]. Bulam et al. [79] reported that a *L. sulphureus* extract contained more than three times more phenolic compounds and flavonoids than a *Trametes versicolor* extract (272.7 mg GAE/g and 44.3 mg QE/mg, respectively, in *L. sulphureus*, and 77.4 mg GAE/ g and 13.8 QE/mg, respectively, in *T. versicolor*). In the study of Nicolcioiu and colleagues [89], the total phenolic content in *L. sulphureus* was higher only than the amount found in *Hericium coralloides* (111.7 mg GAE/l), while the highest value was found in *Agaricus campestris* (489.2 mg GAE/l). Similarly, a small total of phenolic compounds (10.4 mg GAE/g DW) and total contents of phenolic acids (17.7 µg/g DW) were reported by Sułkowska-Ziaja et al. [56]. In another study, *Fomitopsis pinicola* (114.9 µg/g DW) contained the highest total content of phenolic acids, and *Piptoporus betulinus* did not contain them at all. The highest total of phenolic compounds was found in *Fomitopsis pinicola* (21.9 mg GAE/g DW), and the lowest in *Daedaleopsis confragosa* (6.9 GAE/g DW) [56]. Turfan et al. [31] showed that the total phenolics in *L. sulphureus* was 28.7 mg g^−1^, which was the lowest value among the 15 compared mushroom species (the highest content was found in *Boletus edulis* at 157.4 mg g^−1^). In addition, this mushroom contained few flavonoids (12.8 mg g^−1^; the highest levels were found in *Ganoderma lucidum* 30.7 mg g^−1^, and the lowest in *Pleurotus ostreatus* at 8.6 mg g^−1^).

The antioxidant activity of methanol extracts is also determined by the presence of p-hydroxybenzoic acid and cinnamic acid, but in the research of Petrović et al. [21], a higher antioxidant potential was shown by the polysaccharide extract than the methanol extract. A strong antioxidant effect of intracellular polysaccharides and exopolysaccharides derived from mycelium and filtrate has been demonstrated, and polysaccharides subjected to a prior fermentation process have been considered as a potential raw material for the food industry (for the production of health-promoting functional food) and for the pharmaceutical industry [91]. Zhao et al. [27] also showed that polysaccharides extracted with water or enzymatically from the waste mushroom substrate have antioxidant activity and the ability to reduce free radicals. In addition, lovastine isolated from *L. sulphureus* tissues has an antioxidant activity [92].

The antioxidant activity of this fungus has a beneficial effect on the body, as demonstrated by the example of chickens fed feed produced from fermented mycelium [93].

**Antibacterial and antifungal effects.** The antibacterial activity of the fungus *L. sulphureus* was tested using extracts obtained from fruiting bodies or mycelium using various solvents (aqueous (AeE), acetone (AcE), chloroform (ChE), cyclohexane (CyE), dicholoromethane (DmE), ethanol (EhE), ethyl acetate (EAE), and n-hexane (nHE), hydroalcoholic (HAE), methanol (ME) and pertoleum ether (EPE)), as well as selected ingredients isolated from them (glucans, laetiporin C, and laetiporin D). During the tests, the minimum concentration of inhibition and the diameters of the inhibition zone were determined.

The microorganisms used in the research belonged to both the G+ and G− groups of bacteria (mainly pathogenic to humans or causing spoilage of food products). The species whose reaction has been studied most frequently is *Staphylococcus aureus*. The antibacterial activity depended on the species of the tested microorganism, the type of substance used (the type of extract or its components) and the concentration of the tested substance. Inhibition of or reduction in the activity of the tested microorganisms, including the formation of bacterial biofilms [94,95], was most often observed.

AeE, AcE, ChE, CyE, DmE, EhE, EAE, nHE, HAE, ME and EPE *L. sulphureus* extracts reduced the activities of *Escherichia coli*, *Staphylococcus aureus* and *Pseudomonas aeruginosa* [21,33,38,40,80,85,95,96,97,98]. Positive effects of using extracts were also found in the case of *Klebsiella pneumoniae* (EhE, AeE, AcE, EAE and ChE [33]; ME, CyE and DmE [97]) and *Salmonella typhimurium* (AcE, AeE, EhE, ME and DmE [21,40]; ChE and nHE [38]). The Ehe extract also reduced the activities of *Bacillus cereus* [96], *B. subtilis* [85,96], *Enterococcus faecalis* [80,84], *Micrococcus flavus*, *M. luteus*, *Morganella morgani*, *Proteus vulgaris*, *Salmonella enteridis* [96], *Shigella enterica*, *Streptococcus pyogenes* [33], *Staphylococcus epidermidis* [84], and *Yersinia enterecolitica* [96]. The biological activities of *B. subtilis*, *E. faecalis*, *L. monocytogenes*, *M. flavus*, *M. luteus, Sa. abony* and *St. epidermidis* were similarly affected by ME, CyE and DmE extracts of *L. sulphureus* [21,40,97]. ME and DmE extracts effectively reduced the activity of *Helicobacter pylori* [97].

Interesting results were obtained in studies on the effect of α-(1→3)-glucooligosaccharides isolated from *L. sulphureus* tissues on the number and activity of intestinal bacteria (*Bifidobacterium bifidum* ATCC 29521, *B. longum* subsp. *infantis* ATCC 15697, *Lactobacillus acidophilus* DSMZ 20079, *L. acidophilus* PCM 2499, *L. plantarum* ATCC 14917, *L. fermentum* PCM 491, *L. casei* LBY, *L. gallinarum* DSMZ 10532, *L. Johnsonii* DSMZ 10533) and the pathogenic *Escherichia coli* DH5α and *Enterococcus faecalis* PCM 896. Their selective effect and promotion of the increase in the number of desirable microflora in the intestines (microorganisms of the *Bifidobacterium* and *Lactobacillus* genera) and a negative effect on the activity of the tested undesirable bacteria (of the genera *Escherichia* and *Enterococcus*) were found [75].

In studies of antifungal activity, the most frequently used extracts of *L. sulphureus* were AeE, EhE, AcE and ChE, as well as selected components isolated from *L. sulphureus* tissues: polyenes (Seibold et al. 2020 after: [42]), laetiporin C and laetiporin D [60], and lovastine ([92]). The AcE, DmE and ME extracts inhibited the activity of *Aspergillus* fungi (*A. flavus*, *A. fumigatus*, *A. niger*, *A. ochraceus* and *A. versicolor*) [21,33,40]. The AeE, AcE, EAE [33] and ChE [33,38] extracts had a similar effect on *A. fumigatus* and *A. niger*. The nHE extract reduced the growth of *A. fumigatus*, *A. ochraceus* and *A. versicolor* colonies [38]. In addition, the EhE extract decreased the biological activity of *Candida albicans* [33,80,84], *C. paropsilopsis* [84], *C. tropicalis* [80], *Curvularia clavate*, *Geotrichum candidum* and *Fusarium oxysporum* [33], and the AeE, EaE and ChE extracts inhibited the growth of *C. albicans*, *C. clavate* and *G. candidum* mycelia [33]. 

The ME, AcE and DmE extracts also inhibited the activity of fungi pathogenic to plants (*Fulvia fulvum* and *Fusarium sporotrichoides*) and *Penicillium aurantiogriseum* isolated from food [40]. The ME [21], ChE and nHE [38] extracts slowed the growth of mycelia of *Penicillium ochrochloron*, *P. verrucosum* var. *cyclopium* and *Trichoderma viride*. Parvu [99] showed the inhibitory effect of HAE on the development of *Botrytis cinerea*, *F. oxysporum*, *Penicillium gladioli* and *Sclerotinia sclerotiorum*. Laetiporin C also limited the development of *Mucor hiemalis* [60]. 

Sevindik et al. [88] and Sevindik [100] also demonstrated that ME extract has antiviral activity against HSV-1.

**Antitumor effects.** The anticancer effect depends on the type of substance used (the type of extract or component isolated from the fungus), the dose and the tested cancer cell line. Some cancer cells are highly sensitive to substances obtained from *L. sulphureus*. The cytotoxic effect of the test substance is manifested in several ways, such as stopping the cells’ proliferation at various stages of the cell cycle or seriously damaging them (Figure 5). In the case of cancer cells, this is very desirable during the treatment of cancer diseases.

In studies of anticancer activity, the effects of *L. sulphureus* extracts obtained with the use of various solvents have been most often analyzed, and less often the effects of selected isolated components (laetiporin C and laetiporin D [60], sulphureuine B [101], eburicoic acid [69] and lectin [102]). The most frequently studied extract is the ethanol extract, and the most frequently used cancer cell lines are HeLa (human cervical cancer), HCT116 (human colon cancer) and MCF-7 (human breast cancer). The anticancer activity, the migration ability (migration potential), effect on angiogenesis and changes in the levels of selected indicator parameters have been assessed in several studies.

Kim et al. [103] studied the effect of various solvent fractions obtained from methanol extract on the activity of YD-10B cells (human tongue cancer). They found a strong decrease in cell viability under the influence of fractions obtained using hexane and chloroform. Positive results were obtained in the study of ethanol extracts. They caused a strong reduction in the migration potential of HeLa cells and a pro-oxidative effect associated with a significant increase in the level of superoxide anion radicals [104]. Administration of ethanol extract for 45 days at a dose of 250 mg/kg resulted in a decrease in the level of AFP, CEA, MDA, cytochrome b5 and cytochrome P450. In HCC cells (hepatocellular carcinoma), activation of their apoptotic processes was observed, which was associated with a decrease in the expression of Bcl-2 genes and an increase in the expression of mRNA p53, caspases 3 and caspases 9. In general examinations of the condition of the liver tissue, there was a reduction in fat deposition, alleviation of inflammation in hepatocytes and improvement in the condition of the hepatic veins [105]. Similarly, Petrovic et al. [102] found that the ethanol extract inhibited angiogenesis and thus tumor growth of HCT-116 (human colon cancer), but its effect was weaker than that of the lectin isolated from *L. sulphureus*. The lectin also showed a strong anti-migration effect and strongly inhibited tumor metastasis.

Younis et al. [33] proved that aqueous, ethanolic, acetone, ethyl-acetate and chloroform extracts of *L. sulphureus* showed anti-proliferative activity against the cancer cell lines HeLa, HTC 116, MCF-7 and Hep G2 (human liver cancer), but the ethanol extract had the strongest effect on most cells. The aqueous extract also showed a strong antiproliferative effect. Kolundzic et al. [97] obtained a similar effect using cyclohexane and dichloromethane extracts on HeLa and NCI-N87 (gastric carcinoma) cells. Jovanovic et al. [106] found that the ethyl acetate extract changed the activity and inhibited the migration potential of cancer cells from the HCT-116 and HeLa lines, but did not show any cytotoxicity against them. This extract caused a strong increase in ROS and RNS in both cancer cell lines, but the strongest effect was shown against HeLa. In these cells, there was a marked activation of O2- production and a slight increase in GSH levels.

Chepkirui et al. [59] found an anticancer effect of dehydrosulfuric acid and sulfuric acid on A431 (epidermoid carcinoma), A549 (human lung adenocarcinoma), HeLa, MCF-7 and PC-3 (human prostate cancer) tumor cells and the cytotoxic effect of eburicoic acid on HeLa. Similar observations on eburicoic acid were made by He et al. [69]: this component was highly cytotoxic to A-549, HL-60 (human leukemia), SMMC-721 (human hepatocarcinoma) and SW-480 (colorectal cancer) tumor cells. In an earlier study, Lear et al. [57] showed that (±)-laetirobin showed a strong cytostatic effect on HeLa and HCT116 cells: it penetrated them very quickly, blocked divisions in the late stage of mitosis and caused apoptosis. Substances from the laetiporins group isolated from *L. sulphureus* were also tested. Laetiporin A and laetiporin B are cytotoxic to five human cancer cell lines (HeLa, A431, A549, MCF-7, and PC-3) [59], while laetiporin C and laetiporin D have a weak antiproliferative effect against A431, A549, PC-3, and MCF-7 cell lines [60]. Sulphureuine B caused a decrease in Bcl-2 levels and activation of caspase-8, PERK and ATF6α in human glioblastoma cells (U87MG). By changing the activity of mitochondria and receptors, this component inhibited the proliferation of cancer cells and induced their apoptosis [101]. 

The high antioxidant activity of extracts obtained from the fruiting bodies of this fungus indicates that they can protect DNA against damage [80].

**Thrombolytic and anticoagulant effects.** Studies conducted on human blood samples showed a very high (at the level of 69%) thrombolytic activity (TLA) of post-culture liquid left after *L. sulphureus* culture, which is a source of extracellular proteases used in the production of mycopharmaceuticals with TLA [107].

**Anti-inflammatory and immunomodulatory effects.** Sulfurenolids B, C and D isolated from the tissues of the sulphur shelf fungus show anti-inflammatory activity; they have ability to reduce the level of NO in the tissues. A concentration of 50 µM of these substances had a stronger effect than the positive control (minocycline) [61,108]. The fermented ethanol extract also showed an immunomodulatory effect. It improved cell viability (inhibited by LPS), strongly reduced the production of i.a. NO and decreased levels of nuclear factor kappa B (NFkB) and interleukin (IL)-1b. On the other hand, its use favored a slight increase in the levels of TLR4, NFkB and iNOS mRNA [109]. LSL4 lectin (a glycoprotein) has shown a very strong immunomodulatory effect by promoting cell growth and increasing their viability: in vitro application on mouse macrophages resulted in an increase in phagocytic activity and an increase in the levels of NO, iNOS, IL-6, IL-10, IL-1β and TNF -α. According to the authors of the study, LSL4 lectin can be used in medicine (in shaping the activity of the immune system) and in food production (in composing products that meet the requirements of functional food) [110].

**Neuroprotective effects.** Substances contained in *L. sulphureus* make this fungus effective as a neuroprotective agent against diseases associated with the degeneration of the nervous system, for example, Alzeimer’s and Parkinson’s disease [111], according to Ćilerdzić et al., 2018 after: [107].

**Hepatoprotective, gastric analgesic and probiotic roles.** Glycyrrhetinic acid (enoxolone) that is present in the fruiting bodies of *L. sulphureus* effectively relieved stomach pain [35]. Polysaccharides extracted from the mushroom substrate, especially fucose, may find potential use in the prevention of ALD, a liver disease caused by alcohol abuse [27]. Wiater and his colleagues [75] found that alpha (1→3) glucans present in the tissues of sulphur shelf fungus can be used for probiotic purposes. They cause selective stimulation of the growth of the population of bacteria desirable in the human digestive system and negative selection of undesirable microorganisms.

**Insulinogenic and metabolism-modulating effects.** Polysaccharides produced extracellularly by *L. sulphureus* have an insulinogenic effect (Hwang et al., 2008 after [42]), thanks to which they have the ability to regulate cellular metabolism. In addition, the tepenoids (monoterpenes, diterpenes, sesquiterpenes and triterpenes) contained in this mushroom inhibit the activity of alpha-glucosidase. This, in turn, inhibits the formation of monosaccharide molecules and facilitates the formation of glycogen in the liver and muscles [97,112]. 

**Prevention of gynecological problems.** Glycyrrhetinic acid (enoxolone) present in the extract is used to prevent postpartum complications (by accelerating the removal of the placenta after childbirth) [35].

**Dental prophylaxis.** α(1→3) glucans isolated from the cell walls of sulphuric yellow fungus are inducers of mutanases. Their inhibitory effect on the formation of biofilms composed of bacterial cells causing tooth decay has been proven. This creates the possibility of using these substances in dental prophylaxis to remove bacterial biofilms from the surface of the teeth and from prosthetic devices [73].

**Cosmetics effect.** *Laetiporus sulphureus* contains a number of ingredients used in cosmetology. These ingredients have the ability to inhibit the activity of hyaluronidase and tyrosinase. In addition, they have, among others, astringent, UV protective, exfoliating, greasing, moisturizing, healing (sterols), pigmenting (indole derivatives) and whitening (kojic acid) properties [47]. The extract can be used topically to treat skin hyperpigmentation [113]. 

In summary, in recent years, the *Laetiporus sulphureus* has been subjected to intensive research to determine whether it can be used in the prevention and treatment of various diseases (Table 3). The review of the literature provides evidence that extracts of *L. sulphureus* and its selected components have great potential for many branches of medicine. Their antibacterial and antioxidant activity was found. Therapeutic effects have also been demonstrated in relation to civilization diseases related to the functioning of the circulatory system or cancer, neurodegenerative or metabolic diseases. Due to such a wide range of potential effects, this fungus should be used as a raw material for the production of medicines and dietary supplements.

## 5. Fruiting bodies of *Laetiporus sulphureus* in Food Production

*Laetiporus sulphureus* fruiting bodies are edible but valued as traditional food only in some places of occurrence [8,30]. Young fruiting bodies are soft and brittle [24]; hence, in some parts of the world, they are called chicken polypores [2]. Older ones are not suitable for consumption due to the fact that they become hard and difficult to digest [24]. 

Young fruiting bodies have a taste similar to tofu or turkey meat [63], but if the fruit body is not cooked before proper processing or if it is too ripe, then it has a sour taste and smell [24,63]. It is believed that this mushroom is not very aromatic; it has only a slight mushroom aroma. Due to its visual qualities, a yellow color that persists even after heat treatment, it is willingly added to dishes in which it is an interesting colorful ingredient [24]. This mushroom is of culinary importance only locally, e.g., in Germany, the USA or Romania [2]. In many countries (including Poland) its popularity is growing, although people often treat it as a kitchen experiment rather than as a regular ingredient of dinner dishes. On the other hand, in Italy this mushroom is not eaten.

Sulphur shelf fruit bodies can be a substitute for meat in a vegetarian diet. They are prepared in many different ways: they can be fried similar to pork chops (in breadcrumbs) or schnitzels, they can be baked after having been cut into narrow strips, stewed, baked in the oven, deep fried (similar to French fries) or marinated in spices (e.g., honey-spices). In addition, tripe soup, stew (similar to chicken stew), paprikash, and pâté are prepared from this mushroom. They can also be added to soups as an alternative to meat [2,63]. 

Occasionally, mild indigestion may occur after ingestion of *L. sulphureus* fruiting bodies [2,119,120,121], while other reactions have been reported very rarely: allergy, hives, dizziness, intestinal cramps, muscle spasms, vomiting, weakness or anaphylactic shock [119]. However, the symptoms of poisoning have been associated with the consumption of raw or undercooked mushroom [2]. Hence, Szczepkowski [63] recommended pre-scalding or short (2 min) cooking of these mushrooms, and then proper thermal processing, and Mortimer et al. [7] drew attention to the need for long thermal treatment (cooking), which greatly facilitates their digestion. The origin of the fruiting bodies is very important: only those that grow on deciduous trees should be collected, and collection from conifers, such as yew (*Taxus*), should be avoided. In the literature, one can find information that fruiting bodies of *L. sulphureus* collected from conifers may contain toxins present in these trees (Evans 1996 after [2,122,123]).

Thanks to its interesting chemical composition and positive impacts on the human health, this fungus was included in the list of species allowed for marketing or production of mushroom preserves and foodstuffs containing mushrooms in Poland (Regulation of the Min. Health, Journal of Laws of 18 November 2022, item 2365 [124]); however, it was only included in the list in November 2022. This inclusion makes it possible to introduce it on a large scale to the processing and food industry. It is also an opportunity to create products with an original composition that are rich in new bioactive ingredients and have new sensory values. It also opens up new prospects for mushroom growing companies.

**Role in food preservation.** Substances contained in the fruit bodies of *L. sulphureus* prevent the development of microorganisms that cause food spoilage. Volatile compounds extracted from fruiting bodies with various solvents (acetone, methanol and dichloromethane) were tested. All the extracts showed antimicrobial activity against the tested bacteria and fungi [21,40]; moreover, the extract with the strongest effect (methanol extract) clearly inhibited the development of *Aspergillus flavus* on the medium containing tomato puree [40] and on the medium containing poultry meat paste [21]. *Aspergillus flavus* is a microorganism toxic to humans and animals that has the ability to colonize various food products. Its harmfulness is associated with the production of aflatoxins [125,126,127]. Strong antibacterial and antifungal, and even antioxidant effects were associated with substances present in the polysaccharide extract [21]. The authors of the study suggest that extracts from the fruiting bodies of *L. sulphureus* may be a potential natural food preservative [21,40], which is also confirmed by the results of other studies [8,79]. 

## 6. Other Applications

*Laetiporus sulphureus* can be used in the food industry. It can be a source of antioxidants in functional foods, especially when used in the form of a hot alkaline extract [49]. In addition, it can also be an alternative to artificial food preservatives. Its inhibitory effect on the development of the fungus *Aspergillus flavus*, which causes the spoilage of tomato purees and concentrates, has been demonstrated [40]. 

One of the components isolated from this fungus, letiporic acid from the group of polyenes, is used in the clothing and textile industry as a non-toxic, natural dye for silk. It gives fabrics an orange-yellow color [42]. Sulphur shelf fungus is also used for the production of biopolymers, which can then be a raw material for various industries [52]. 

This fungus can be used in environmental protection, especially in sewage treatment. For this purpose, its ability to synthesize extracellular enzymes, such as manganese oxidase and linnocellulite enzymes, including lignin peroxidase, can be used [94,120]. *Laetiporus sulphureus* enzymatically decomposes waste and impurities containing lignin, hemicellulose and cellulose, and as a result, soluble compounds of low molecular weight are formed (Lim et al., 2013 after: [89]).

## 7. Conclusions

Recognizing the fungus *L. sulphureus* as a useful species and allowing it to be used in food processing and production is a challenge and, at the same time, a great opportunity for many areas of the economy. In the case of areas with a low degree of urbanization, including rural areas, it is an opportunity to create new jobs or find an additional source of income for residents. People with various degrees of education, including unskilled workers, can be employed in the cultivation of this mushroom. This gives them a great opportunity to raise their economic status. Cultivation of this mushroom is relatively simple (requiring only compliance with established rules) and does not require the operation of technologically advanced equipment.

Sulphur shelf fungus is a relatively cheap raw material because its cultivation can be carried out on a substrate consisting of cheap and easily available components. The components of the substrate may be, for example, the remains of crop plants from which biogas was previously obtained [15]; in such cases, cultivation is a form of agricultural waste management. The cultivation process does not require fertilization or heavy irrigation and there are no special thermal requirements related to the need to maintain a constant temperature, which results in additional cost reductions. Interestingly, a slight supercooling of the culture does not have a negative effect; on the contrary, it initiates the formation of fruiting bodies [14]. Sulphur shelf mycelium is widely available, and its propagation is not laborious and does not generate additional costs. This mushroom can be cultivated continuously (all year round), especially in mild winter climates. It is characterized by rapid growth. Less than 2 months after the inoculation of the substrate with mycelium, it forms numerous fruiting bodies of large weight [13].

*Laetiporus sulphureus* fruiting bodies are a relatively new raw material for the food industry. Due to their chemical composition, they are a rich source of fiber and many other bioactive ingredients. Many of the chemicals found in this mushroom can also be extracted. Studies have shown that some of the valuable ingredients are included in water extracts. This means that dishes prepared with *L. sulphureus* with the use of water (soups, sauces, stewed dishes) will have a health-promoting effects. Due to the protein content, this mushroom can be a partial or complete substitute for meat. It can be an interesting addition to meat products such as pâtés or sausages, vegetable spreads for sandwiches, bigos, letchos or fillings for tarts or stuffing for dumplings. It can be used in a cooked or dried form, in small pieces or in a highly fragmented (ground) form. Dried ground fruiting bodies can be a nutrient-rich addition to meat or vegetable dishes. *Laetiporus sulphureus* fruiting bodies can also be used to obtain natural chemicals, which can then be used in medicine and in the production of dietary supplements.

The color abstract features a photo by Congerdesign (source: Pixabay).

## Figures and Tables

**Figure 1 foods-12-01539-f001:**
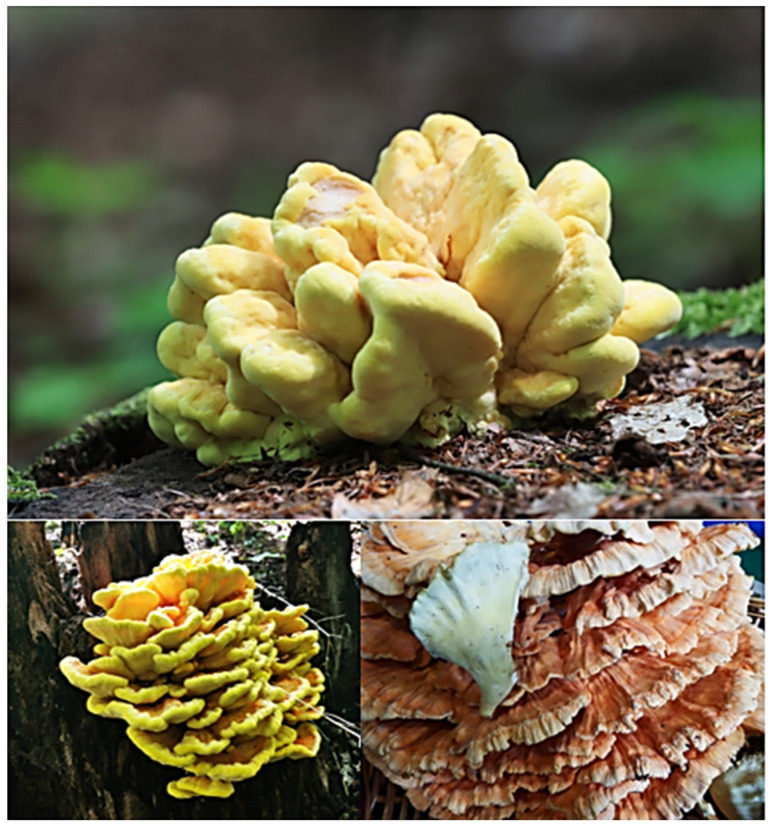
Fruiting bodies of *Laetiporus sulphureus* in different age stages: young (at the **top**), mature (**bottom left**) and aging fruit body (**bottom right**). Authors of the photos: Andrzej Chruślak (**top** and **bottom left**) and Pierino Bigoni (**bottom right**).

**Figure 2 foods-12-01539-f002:**
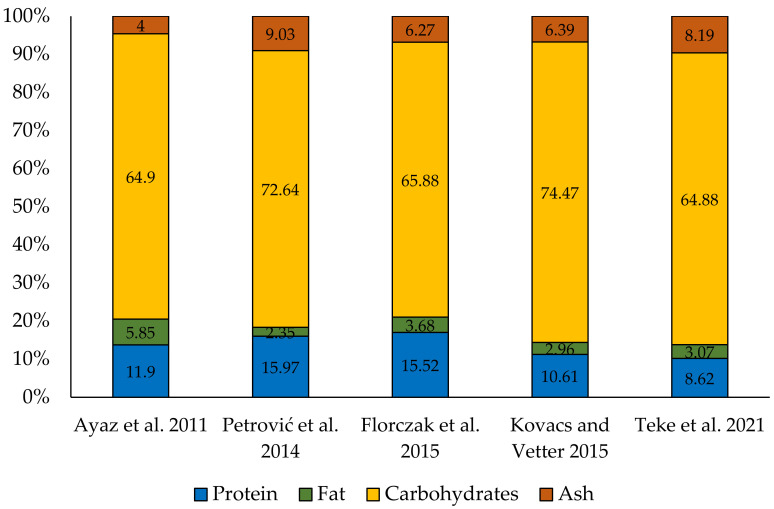
Nutritional value of fruiting bodies of *Laetiporus sulphureus* (% dw). Cited for: Ayaz et al. [20], Petrović et al. [21], Florczak et al. [24], Kovacs and Vetter [1] and Teke et al. [22].

**Figure 3 foods-12-01539-f003:**
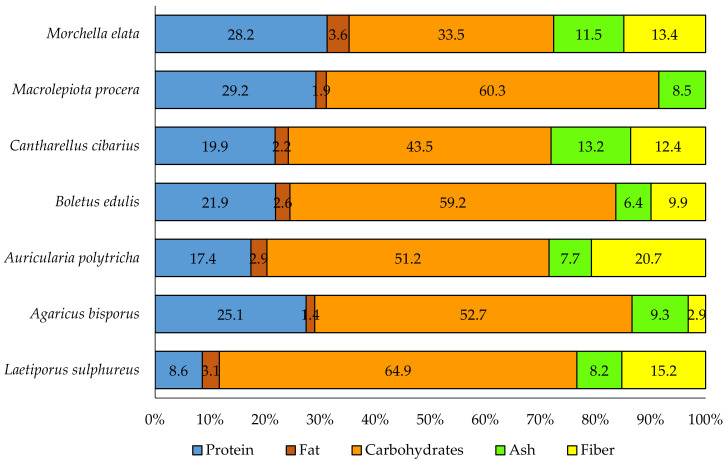
Comparison of selected nutrients in selected species of mushrooms (data on *Agaricus bisporus*, *Boletus edulis*, *Cantharellus cibarius* and *Morchella elata*—TEI and University of Thessaly, after: [76], *Auricularia polytricha* and *Laetiporus sulphureus* [77], *Macrolepiota procera* [78]).

**Figure 4 foods-12-01539-f004:**
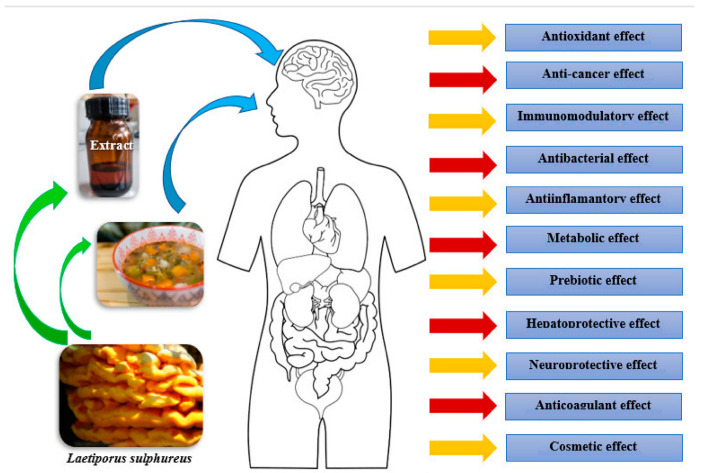
Bioactivity of *Laetiporus sulphureus.* Image source: Pixabay; authors of images: Amberrose Nelson, Kevin and Rihajin. Modified by: Iwona Adamska.

**Figure 5 foods-12-01539-f005:**
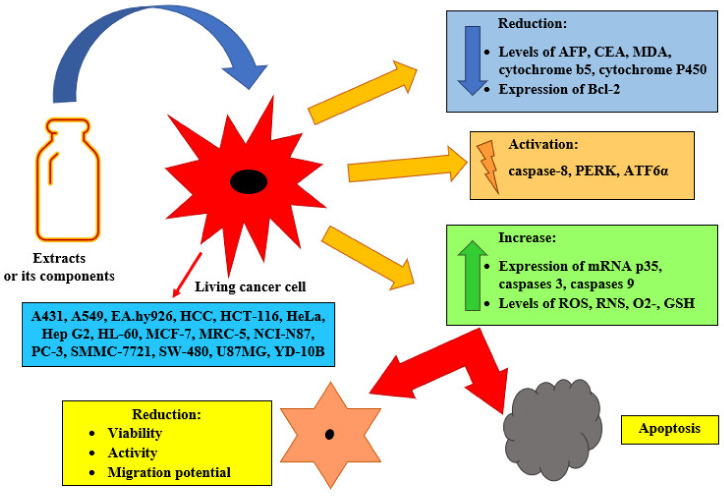
The effects of substances contained in *Laetiporus sulphureus* on cancer cells.

**Table 1 foods-12-01539-t001:** The richness of chemical components found in *Laetiporus sulphureus*.

Component	Source of Information about Presence of Component	Chemical Group	Molecular Formula	Molecular Weight (g/mol)	Source of Information about Properties of Component
Carbohydrates and their derivatives					
Arabinose	[26,27]	monosaccharides	C_5_H_10_O_5_	150.13	[28,29]
Fructose	[30]		C_6_H_12_O_6_	180.16	[28,29]
Galactose	[27]		C_6_H_12_O_6_	180.16	[28,29]
Glucose	[8,26,27,31]		C_6_H_12_O_6_	180.16	[28,29]
Mannose	[26,27]		C_6_H_12_O_6_	180.16	[28,29]
Xylose	[26,27]		C_5_H_10_O_5_	150.13	[28,29]
Fucose	[26,27]	deoxy sugars	C_6_H_12_O_5_	164.16	[28,29]
Rhamnose	[26]		C_6_H_12_O_5_	164.16	[28,29]
Sucrose	[8,31]	disaccharides	C_12_H_22_O_11_	342.30	[28,29]
Trehalose	[14,30]		C_12_H_22_O_11_	342.30	[28,29]
Laminaran	[26]	polysaccharides	C_18_H_32_O_16_	504.44	[28,29]
Mannitol	[21,30]	sugar alcohols (polyol)	C_6_H_14_O_6_	182.17	[28,29]
Matsutakic acid (masutakic acid)	[32]	acetylenic acids	C_10_H_16_O_4_	200.23	[29]
Egonol glucoside	Yoshikawa et al., 2001 after: [33]	glucosides	C_25_H_28_O_10_	488.48	[28,29]
Lipids					
Linoleic acid	[21,30,34,35,36,37,38]	unsaturated fatty acids	C_18_H_32_O_2_	280.45	[28,29]
Oleic acid	[21,30,33,34,35,36,37,38,39]		C_18_H_34_O_2_	282.46	[28,29]
Palmitoleic acid	[34]		C_16_H_30_O_2_	254.41	[28,29]
Isovaleric acid	[40]	saturated fatty acids	C_5_H_10_O_2_	102.13	[28,29]
Myristic acid	[34]		C_14_H_28_O_2_	228.37	[28,29]
Palmitic acid	[21,30,34,35,36,37,38]		C_16_H_32_O_2_	256.42	[28,29]
Stearic acid	[34,37]		C_18_H_36_O_2_	284.48	[28,29]
9-Octadecenoic acid (Z)-, 2-hydroxy-1-(hydroxymethyl)ethyl ester; (2-Oleoylglycerol)	[33]	glycerolipids	C_21_H_40_O_4_	356.54	[28]
Ethyl cholate	[33]	steroids	C_26_H_44_O_5_	436.63	[28]
Laetiporic acid	[41,42]	polyenes	C_27_H_32_O_4_	420.50	[28,29]
2-dehydro-3-deoxylaetiporic acid A	[43]		C_27_H_30_O_3_	402.53	[28,29]
6-((2E, 6E)-3, 7-dimethyldeca-2, 6-dienyl)-7-hydroxy-5-methoxy-4-methylphtanlan-1-one	[44]		C_22_H_30_O_5_	374.21	[44]
Amino acids and peptides					
Cysteine	[45]	exogenous amino acids	C_3_H_7_NO_2_S	121.16	[28,29]
Ergothioneine	[46]		C_9_H_15_N_3_O_2_S	229.30	[28,29]
L-Histidine	[30,36]		C_6_H_9_N_3_O_2_	155.15	[28,29]
Isoleucine	[30,36]		C_6_H_13_NO_2_	131.17	[28,29]
Leucine	[30,36]		C_6_H_13_NO_2_	131.17	[28,29]
Lysine	[30,45]		C_6_H_14_N_2_O_2_	146.19	[28,29]
Methionine	[30,36]		C_5_H_11_NO_2_S	149.21	[28,29]
Phenylalanine	[45]		C_9_H_11_NO_2_	165.19	[28,29]
Threonine	[30,36]		C_4_H_9_NO_3_	119.12	[28,29]
Tryptophan	[36,47]		C_11_H_12_N_2_O_2_	204.23	[28,29]
Valine	[45]		C_5_H_11_NO_2_	117.15	[28,29]
5-hydroxy-L-tryptophan	[47]		C_11_H_12_N_2_O_3_	220.22	[28,29,48]
Alanine	[45]	endogenous amino acids	C_3_H_7_NO_2_	89.09	[28,29]
Arginine	[36]		C_6_H_14_N_4_O_2_	174.20	[28,29]
Aspartic acid	[45]		C_4_H_7_NO_4_	133.10	[28,29]
Glycine	[45]		C₂H₅NO₂	75.07	[28,29]
Proline	[45]		C_5_H_9_NO_2_	115.13	[28,29]
Serine	[45]		C_3_H_7_NO_3_	105.09	[28,29]
Beauvericin	Del et al., 1978 after: [32]	peptides	C_45_H_57_N_3_O_9_	783.95	[28,29,48]
Carboxylic acids					
Ascorbic acid	[32,46,49,50]		C_6_H_8_O_6_	176.13	[28,29]
Cinnamic acid	[3,21]		C_9_H_8_O_2_	148.16	[28,29]
Citric acid	[3,20,21,32,51]		C_6_H_8_O_7_	192.12	[28,29]
Enoxolone (glycyrrhetinic acid)	[36]		C_30_H_46_O_4_	470.68	[28,29,48]
Fumaric acid	[21]		C_4_H_4_O_4_	116.07	[28,29]
Lactic acid (polylactic acid PLA)	[52]		C_3_H_6_O_3_	90.08	[28]
Malic acid	[20,32]		C_4_H_6_O_5_	134.09	[28,29]
Malonic acid	[51]		C_3_H_4_O_4_	104.06	[28,29]
Oxalic acid	[3,21]		C_2_H_2_O_4_	90.03	[28,29]
Quinic acid	[21]		C_7_H_12_O_6_	192.17	[28,29]
Tartaric acid	[51]		C_4_H_6_O_6_	150.09	[28,29]
Vitamins					
Cholecalciferol	[30]	D_3_	C_27_H_44_O	384.64	[28,29]
Cyanocobalamin	[8]	B_12_	C₆₃H₈₈CoN₁₄O₁₄P	1 355.38	[29]
ʆ-tocopherol	[3,30,46,49]	E	C_29_H_50_O_2_	430.71	[28,29]
β-tocopherol	[3,30,46]		C_28_H_48_O_2_	416.68	[28,29]
ɣ-tocopherol	[3,46]		C_28_H_48_O_2_	416.68	[28,29]
Caffeic acid	[53]	Phenolic acids	C_9_H_8_O_4_	180.16	[28,29]
Chlorogenic acid	[53]		C_16_H_18_O_9_	354.31	[28,29]
p-Coumaric acid	[53,54]		C_9_H_8_O_3_	164.16	[28,29]
Gallic acid	[47,53,55]		C_7_H_6_O_5_	170.12	[28,29]
4-Hydroxybenzoic acid	[3,21,47,54]		C_7_H_6_O_3_	138.12	[28,48]
Kojic acid	[47]		C_6_H_6_O_4_	142.11	[28,29]
Protocatechuic acid	[47,54,55,56]		C_7_H_6_O_4_	154.12	[28]
Salicylic acid	[54]		C_7_H_6_O_3_	138.12	[28]
Egonol	Yoshikawa et al., 2001 after: [33]	Benzofurans and derivates	C_19_H_18_O_5_	326.34	[28,29]
Demethoxyegonol	Yoshikawa et al., 2001 after: [33]		C_18_H_16_O_4_	296.32	[28,29]
Egonol gentiobioside	Yoshikawa et al., 2001 after: [33]		C_31_H_38_O_15_	650.6	[29]
Matsutakeside I	[32]		C_30_H_36_O_14_	620.60	[29]
(±)-Laetirobin (Laetiporina)	[57]		C_44_H_32_O_12_	752.72	[28,29]
Ergosterol (Provitamin D2)	[46]	Sterols	C_28_H44O	396.65	[28,29]
Dehydroergosterol	[58]		C_28_H_42_O	394.63	[28,29]
Ergost-7-en-3-ol	[58]		C_28_H_48_O	400.68	[28,29]
Ergost-3,5,7,9(11),22-pentaen	[58]		C_28_H_40_	376.60	[29]
24-methylenelanost-8-en-3-ol (obtusifoldienol)	[58]		C_31_H_52_O	440.74	[28,29]
4,4-dimethylergost-24-en-3-ol	[58]		C_30_H_52_O	428.70	[29]
4-methylergost-5,7,25-trien-3-ol	[58]		C_29_H_46_O	410.70	[29]
4-methylergost-7,14,25-trien-3-ol	[58]		C_29_H_46_O	410.70	[29]
Ergosterol peroxide	[58]		C_28_H_44_O_3_	428.65	[28,29]
Ergosta-7,22-dien-3,5,6- triol (cerevisterol)	[58]		C_28_H_46_O_3_	430.66	[28,29]
Laetiporin A	[59]	Triterpenoids	C_31_H_49_O_3_	469.37	[59]
Laetiporin B	[59]		C_34_H_53_O_7_	573.38	[59]
Laetiporin C	[60]		C_31_H_50_NaO_5_	525.35	[60]
Laetiporin D	[60]		C_31_H_48_NaO_5_	523.34	[60]
Fomefficinic acid	[60]		C_31_H_48_O_4_	484.70	[29]
Eburicoic acid	[60,61]		C_31_H_50_O_3_	470.73	[29,61]
Dehydroeburicoic acid	[59]		C_31_H_48_O_3_	468.71	[28,29]
15 α-hydroxytrametenolic acid	[60]		C_30_H_48_O_4_	472.7	[29]
Trametenolic acid	[60]		C_30_H_48_O_3_	456.70	[28,62]
Sulphurenic acid	[63]		C_31_H_50_O_4_	486.73	[28,29]
Sulphurenoid A	[61]		C_27_H_42_O_5_	445.30	[61]
Sulphurenoid B	[61]		C_27_H_40_O_5_	443.27	[61]
Sulphurenoid C	[61]		C_27_H_44_O_5_	447.31	[61]
Sulphurenoid D	[61]		C_30_H_44_O4	467.32	[61]
15α-hydroxy-3-oxolanosta-8,24-dien-21-oic acid	[61]		C_30_H_46_O_3_	454.68	[64]
3-keto-dehydrosulfurenic acid	[61]		C_31_H_46_O_4_	481.3	[65]
3-oxolanosta-8,24-dien-21- oic acid (pinicolic acid A)	[61]		C_30_H_46_O_3_	454.70	[29,66]
5α-hydroxytrametenolic acid	[61]		C_30_H_48_O_4_	472.7	[63]
3- oxosulfurenic acid	[61]		C_31_H_48_O_4_	484.7	[66]
Dehydrosulphurenic acid	[61]		C_31_H_48_O_4_	484.7	[66]
Acetyl eburicoic acid (LSM-H7)	Leon et al., 2004 after: [33,67]		C_33_H_52_O_4_	512.8	[29,67]
Acetyl trametenolic acid	León et al., 2004 after: [68]		C_32_H_50_O_4_	498.70	[29]
Versisponic acid A	Yoshikawa et al., 2000 after: [68]		C_30_H_48_O_5_	488.70	[28,29]
Versisponic acid B	Yoshikawa et al., 2000 after: [68]		C_32_H_48_O_5_	512.72	[28,29]
Versisponic acid C	Yoshikawa et al., 2000 after: [68]		C_33_H_50_O_5_	526.75	[28,29]
Versisponic acid D	Yoshikawa et al., 2000 after: [68]		C_33_H_52_O_5_	528.76	[28,29]
Versisponic acid E	Yoshikawa et al., 2000 after: [68]		C_35_H_54_O_5_	554.80	[28]
3β-hydroxylanosta-8,24-dien-21-oic acid	[61]		C_30_H_48_O_3_	456.70	[28]
laricinolic acid	[61]	Sesquiterpenoids	C_15_H_24_O_3_	252.35	[28]
Sulphureuine A	[69]		C_15_H_22_O_2_	234.33	[28,29,69]
Sulphureuine B	[69]		C_15_H_28_O_4_	272.20	[69]
Sulphureuine C	[69]		C_15_H_28_O_4_	272.20	[69]
Sulphureuine D	[69]		C_15_H_26_O_3_	254.19	[69]
Sulphureuine E	[69]		C_15_H_24_O_4_	268.17	[69]
Sulphureuine F	[69]		C_15_H_24_O_3_	252.17	[69]
Sulphureuine G	[69]		C_15_H_28_O_3_	256.20	[69]
Sulphureuine H	[69]		C_15_H_26_O_3_	254.18	[69]
Agripilol A	[69]		C_15_H_28_O_4_	272.38	[28,29]

**Table 2 foods-12-01539-t002:** Content of selected minerals in *L. sulphureus* fruiting bodies.

	Bulam et al. [79] (mg/kg dw)	Teke et al. [22] (mg/kg^−1^ dw)	Bengu [34] (mg/kg^−1^ dw)	Sevindik et al. [80] (mg/kg^−1^ dw)	Turfan et al. [31] (mg/kg^−1^ dw)	Florczak et al. [24] (mg/ kg^−1^ dw)	Kovacs and Vetter [1] (mg/kg^−1^ dw)	Luangharn et al. [25] (mg/kg dw)
Ca	0.49 ± 0.01	13.04 ± 0.11	nd	nd	18.78 ± 0.06	1.02 ± 0.77	765 ± 55.1	2.59 ± 0.01
K	nd	433.62 ± 4.28	nd	nd	5752.54 ± 8.32	nd	28,940 ± 2174	nd
Mg	4.59 ± 0.01	13.85 ± 0.79	nd	nd	16.86 ± 0.90	2.90 ± 0.45	1001 ± 15.5	1.09
P	24.52 ± 0.09	542.88 ± 4.26	nd	nd	1524.50 ± 4.32	nd	4890 ± 575	nd
Na	1.88 ± 0.01	4.20 ± 0.58	nd	nd	8.00 ± 0.30	nd	209.9 ± 141.0	11.01 ± 0.12
Cu	0.04 ± 0.001	1.15 ± 0.06	5.00	1.90 ± 1.22	14.35 ± 0.11	0.52 ± 0.093	9.72 ± 4.90	0.14 ± 0.01
Fe	0.49 ± 0.02	8.69 ± 0.46	162.92	138.44 ± 21.22	80.62 ± 0.54	2.88 ± 0.12	50.9 ± 17.30	2.28 ± 0.03
Zn	0.21 ± 0.001	2.66 ± 0.18	28.360	47.42 ± 6.60	113.63 ± 9.47	0.22 ± 0.012	56.5 ± 6.10	1.20
Al	0.89 ± 0.01	nd	nd	nd	27.96 ± 0.18	nd	34.57 ± 23.00	nd
Ba	0.07 ± 0.09	nd	nd	nd	nd	nd	3.04 ± 1.89	nd
As	0.01 ± 0.001	nd	nd	nd	2.04 ± 0.02	nd	bd	nd
Mn	0.03 ± 0.001	nd	19.360	nd	99.49 ± 0.41	nd	5.18 ± 1.10	0.35 ± 0.02
B	0.04 ± 0.001	nd	nd	nd	nd	nd	nd	nd
Co	0.001 ± 0.001	nd	nd	nd	1.76 ± 0.40	nd	0.33 ± 0.13	nd
Cd	0.003 ± 0.001	nd	nd	nd	0.41 ± 0.02	nd	1.79 ± 2.00	nd
Pb	0.004 ± 0.001	nd	nd	1.73 ± 0.89	2.45 ± 0.01	nd	nd	nd
Ni	0.042 ± 0.001	nd	nd	0.00 ± 00	9.54 ± 0.15	nd	1.36 ± 0.69	nd
Cr	0.008 ± 0.001	nd	nd	nd	4.03 ± 0.03	nd	0.55 ± 0.07	nd

Symbols: nd—no data; bd—below detection.

**Table 3 foods-12-01539-t003:** Research on the activity of *Laetiporus sulphureus*.

Body System (or Part)	Documented Effect	Source of Information
Protection: 1. Anti-aging effect	Antioxidant effect	[1,21,31,56,79,80,84,85,88,89,90,91,92,93]
2. Anti-cancer effect	Cytotoxic Anti-proliferating Protective for DNA	[24,33,49,56,57,60,80,97,102,103,104,105,106,114]
Immunity	Anti-inflammatory Antibacterial Antifungal Antiviral Antimalarial Immunomodulating	[24,33,38,40,54,60,63,80,94,95,96,97,98,100,111,115,116,117], Seibold et al., 2020 after: [42]
Metabolism	Hypoglycemic Hipolipemic Insulinogenic	[49,112], Hwang et al., 2008 after: [42]
Digestive system	Prebiotic (improves digestion) Antiulcer Hepatoprotective Relieving stomach pains	[35,75]
Circulatory system	Antithrombin Anticoagulant Haemolyzing	[114]
Nervous system	Acetylocholinesterase inhibiting Antidepressant and neuroprotective Prevention of Alzheimer’s and Parkinson’s diseases	[107,111]
Reproductive system	Reducing the incidence of postpartum problems in women	[35]
Skin	Healing Whitening Pigmenting Exfoliating epidermis Moisturizing Protective Anti-sweat Astringent Inhibition of melanogenesis	[47,107]
Dental prophylaxis	Elimination of bacterial biofilms causing tooth decay	[73]
Possible industrial applications		
Food technology	Food preservative	[8,21,40,79]
Material industry	Production of biopolymer composites	[52]
Clothing and textile industry	Yellow dye for dyeing textiles	[42,63]
Chemical industry (production of agents used to protect the environment)	Biological preparations accelerating the decomposition of cellulose impurities	[118,119], Lim et al. after: [90]

## Data Availability

Not applicable.

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
