# Peer review of "The Possibility of Using Sulphur Shelf Fungus (Laetiporus sulphureus) in the Food Industry and in Medicine—A Review"

_foods, 2023, doi:10.3390/foods12071539_

Round 1

Reviewer 1 Report

1-The word "fungus" is missing in the title. (Sulphur Shelf Fungus – Laetiporus sulphureus)

2-In the title, it could be stated that the study was a compilation.

3-The last sentence of the summary could have been the title. (Line 12-14)

4-Likewise mushrooms could be added to keywords.

5-While the data range is given as 2016-2023 in the summary, it is given as 2015-2023 in the introductory section on line 26.

6-There was a line shift between 34-35.

7-There are numbering errors.

-1. Introduction (Line 18)

-1. Characteristics and occurrence of Laetiporus sulphureus (Line 29)

-1. Nutritional value of fruiting bodies (Line 67)

-1. Bioactivity of Laetiporus sulphureus (Line 274)

-1. Fruitbodies of Laetiporus sulphureus in food production (line 479)

-1. Other application (line 536)

-1. Conclusions (line 552)

8-In the 338th row, antibacterial effect was mentioned, but the table given is antimicrobial. Again, it was called antifungal in the antibacterial title (line 372).

9-Similarly, the title of antitumor and antimutagenic is given in line 379, but the table and content are in the form of anticancer activity.

10-In general, there are problems in the titles, apart from that, it has been a useful study that will contribute to the literature.

Author Response

Dear Reviewer,

Thank you very much for all your comments in the review of my manuscript. I am convinced that they increase the value of my publication. I followed all the instructions and made the necessary corrections in the publication.

List of changes made:

  1. The title of the work was changed to suggested (i.e. the last sentence in the abstract), adding information that the publication is a literature review.
  2. Added the Latin name of the fungus to the keywords and supplemented the common name with the word "fungus"
  3. The range of years from which the data was analyzed was corrected so that the information in the abstract and in the introduction are consistent (2016-2023)
  4. Editorial corrections of the entire work were made (shifts were removed and the font size was adjusted). These errors resulted from sending the thesis in an accelerated mode, i.e. without using the MDPI template. Editing errors occurred after the manuscript was automatically pasted into the template. I am very sorry, but I did not see the version in the template that was sent for review and I did not know that such errors would arise (no chapter numbering, non-uniform type and size of line shift font).
  5. The numbering of chapters has been corrected to the correct one.
  6. The nomenclature of the chapters and the tables included in them have been unified and made more precise, so that the content of the table corresponds to the content of the chapter.

Thank you very much for taking the time to read and correct my work.

Best regards,

Iwona Adamska

Reviewer 2 Report

The manuscript appears to be poorly organized and confusing, with numerous inconsistencies and unclear sections that make it difficult for the reader to understand the main points. The excessive use of tables, which comprise nearly half of the manuscript, is unnecessary and detracts from the overall flow of the content. Additionally, the structure of the compounds in Table 1 is ambiguous and unprofessional, which is unacceptable in a manuscript of this caliber. The lack of cohesion and clarity in the manuscript is highly disappointing and reflects poorly on the author's attention to detail and professionalism. While it is commendable that the manuscript aims to shed light on the potential value of Laetiporus sulphureus, the lack of clarity and organization undermines the significance of the topic.

  The manuscript is encouraged to resubmit after thorough revision.

major concerns

1.      While the chemical structures in Table 1 are informative, they are not presented in a uniform format. Please ask the authors to redraw the structures in a uniform format to improve readability and presentation.

2.      The title of Table 1 is "The main components of the L. sulphureus," but the contents of the table are not consistent with the title. If the authors wish to reflect the richness of the components, then the title of Table 1 should be changed, and the small molecule compounds of Laetiporus sulphureus origin should be listed as far as possible. The authors should consider listing small molecules as a separate category, as they have nutritional value. Moreover, the grouping of these small molecules should be organized by compound type, with those of the same type placed together. Please refer to the article "Bioactive components of Laetiporus species and their pharmacological effects, 10.1007/s00253-022-12149-w" for examples.

3.      The organization of the active section is not clear, and the summary concluding description should have been given first, rather than a large pile of examples. The authors should reorganize this section to improve clarity and coherence.

4.      The domestication and artificial cultivation of L. sulphureus should be mentioned and discussed. This information is important for readers interested in the practical applications of L. sulphureus.

5.      Genome sequencing has greatly facilitated the research and development of medicinal edible mushrooms. As a review aimed at synthesizing the edible and medicinal values of L. sulphureus, a summary and discussion of the current status of their genomic research is essential. Please refer to and cite the following articles: "The genome sequence of the chicken of the woods fungus, Laetiporus sulphureus (Bull.) Murrill, 1920, 10.12688/wellcomeopenres.17750.1" and "Chromosome-Level Genome Sequences, Comparative Genomic Analyses, and Secondary-Metabolite Biosynthesis Evaluation of the Medicinal Edible Mushroom Laetiporus sulphureus, 10.1128/spectrum.02439-22".

Minor errors

The manuscript contains many errors and deficiencies in detail, including but not limited to the following.

1.      Inconsistent font size within a paragraph is outrageous, however, I have encountered it. May I ask the author if you don't preview your manuscript after submission? Which are in line 301-334.

2.      Candida albicans in 367.

3.      The page numbers of the references are incomplete and the DOI is incomplete, please check each one. 

Author Response

Dear Reviewer,

Thank you very much for all your comments in the review of my manuscript. I am convinced that they increase the value of my publication. I followed the instructions and made the necessary corrections in the publication.

List of changes:

  1. Table 1 has been rebuilt. The structure of the compounds was removed from it, and instead it was expanded to include a number of biologically active compounds listed in the publication indicated by the reviewer and grouped according to the rules adopted in this work. The title of the table was also changed to the one suggested by the Reviewer (The richness of components contained in Laetiporus sulphureus).
  2. The manuscript was adapted to the requirements of the MDPI publishing house related to the placement of tables immediately after the paragraph in which they were cited, hence they are in the middle of the text. This may give the impression of text tearing in a manuscript, but in published works, they appear as small interactive windows that can be opened.
  3. Editorial corrections of the entire work were made (shifts were removed and the font size was adjusted). These errors resulted from sending the thesis in an accelerated mode, i.e. without using the MDPI template. Editing errors occurred after the manuscript was automatically pasted into the template. I am very sorry, but I did not see the version in the template that was sent for review and I did not know that such errors would arise (including non-uniform font type and size, and text shifts).
  4. The chapter on the biological activity of L. sulphureus has been reorganized. The table summarizing this chapter has been moved to the end of the chapter. A short introduction has been added at the beginning of the chapter.
  5. Added information on the possibility of cultivating L. sulphureus in home conditions and in controlled conditions (on various types of nutrient substrates).
  6. A chapter on genome sequencing research has been added. It uses information from the literature indicated by the reviewer.
  7. Lettering errors have been removed from the work.
  8. Corrections (supplements) of page numbers of cited publications were made and all DOI addresses were checked.

Thank you very much for taking the time to read and correct my work.

                                                                                    Best regards,

                                                     Iwona Adamska

Round 2

Reviewer 2 Report

A review exists to summarise information, to condense and distil ideas, not to pile up data. Table III in its existence or current form makes little sense, can't simple antimicrobial information be summarised in words and is it necessary to list the non-resistance of a particular ingredient to certain bacteria? The existence of Tables III and IV is a burden on an already overly long manuscript. Please just summarise carefully and show valid information.

Although the current manuscript has been greatly improved, it is still not acceptable. Please continue to condense the length.

Author Response

West Pomeranian University of Technology in Szczecin                                   Szczecin, 30.03.2022

Department of Fisch, Plant and Gastronomy Technology

Papieża Pawla VI Street No. 3

71-459 Szczecin

Dear Reviewer,

Thank you very much for all your comments in the review of my manuscript. I am convinced that they increase the value of my publication. I followed the instructions and made the necessary corrections in the publication.

List of changes:

  1. Tables III and IV. have been removed from the manuscript and the information contained therein, after analysis, is presented in the form of a descriptive data summary. This significantly shortened the length of the manuscript and made it possible to highlight the relationships between the compiled data. This amendment, in my opinion, has had a very positive effect on the work.
  2. A linguistic revision was carried out.

Thank you very much for taking the time to read my manuscript and point out areas for improvement.

Best regards,

Iwona Adamska
